# Graph Neural Networks with Directional Encodings for Anisotropic Elasticity

## Abstract

Simulating the behavior of nonlinear and anisotropic materials is a central problem with applications across engineering, computer graphics, robotics, and beyond. While conventional mesh-based simulations provide accurate and reliable predictions, their computational overhead typically prevents their use in interactive applications. Graph neural networks (GNN) have recently emerged as a compelling alternative to conventional simulations for time-critical applications. However, existing GNN-based methods cannot distinguish between deformations in different directions and are thus limited to isotropic materials. To address this limitation, we propose a novel and easy-to-implement GNN architecture based on directional encodings of edge features. By preserving directional information during message passing, our method has access to the full state of deformation and can thus model anisotropic materials. We demonstrate through a set of qualitative and quantitative evaluations that our approach outperforms existing mesh-based GNN approaches for modeling anisotropic materials.

## 1 Introduction

From plant leaves to animal muscle and from woven textiles to fiber-reinforced composites—many natural and engineered materials are strongly anisotropic, *i.e.*, their stress response varies significantly depending on the direction of deformation. Simulating such anisotropic materials properties is crucial for many applications in science and engineering (1). Conventional simulation methods typically rely on mesh-based finite element discretizations for numerical solutions of the underlying partial differential equations. While these methods can capture intricate material behavior with high accuracy, they come at a substantial computational cost. Striking a balance between accuracy and efficiency, learning-based methods have emerged as a promising alternative to conventional simulations. Arguably the closest analogy to mesh-based simulation is a mesh-based deep neural representation. Indeed, existing works built on mesh-based graph neural networks (MGNN) have shown promising results (2; 3). While existing MGNN methods have focused on isotropic materials so far, accounting for anisotropy might seem a straightforward extension. Unfortunately, the message passing architectures of current MGNNs rely on spatial averaging of edge features, which discards all directional information on deformation. As we show in our analysis, discarding directional information means that existing MGNNs are unable to model anisotropic materials.

In this work, we present a novel feature encoding scheme designed to preserve directional information during message passing. We decompose edge features into components along three material-space basis vectors and aggregate these components separately during message passing. In this way, feature averaging takes into account the material-space orientation of the edges, leading to significantly improved preservation of anisotropic information. This improvement requires minimal changes to standard mesh-based graph neural networks, thus allowing for easy integration into existing frameworks. We validate our approach on a set of qualitative and quantitative examples and demonstrate that our approach outperforms the state-of-the-art method for capturing material anisotropy.

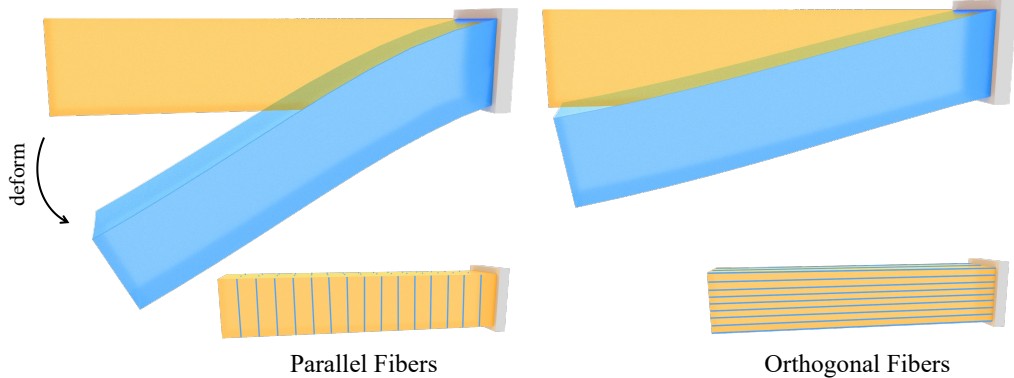

Figure 1: Anisotropic Elasticity. We apply our approach to model the nonlinear deformation of an elastic cantilever beam under gravitational load. The beam is made from an isotropic base material augmented with reinforcing fibers (see insets). On the left, fibers are oriented in parallel to the direction of gravity, which leads to only minor stiffening compared to the base material. On the right, fibers run along the axis of the beam, leading to significantly reduced deflection for this load case. Rest and deformed states are shown in orange and blue, respectively.

## 2 RELATED WORK

**Simulation of Deformable Objects** Simulating deformable objects plays a pivotal role across various disciplines, including mechanical engineering, computer graphics, and robotics. Among existing approaches, which include particle-based (4; 5; 6), grid-based (7; 8; 9; 10) and hybrid methods (11; 12), mesh-based representations are arguably the most prevalent choice (13; 14; 15; 16). The computer graphics community has made great strides in efficient, robust, and accurate mesh-based simulation of deformable bodies (17; 18; 19; 20; 21). Although dimension reduction techniques exist (22; 23; 24; 25), the associated computational costs for native scale simulation are often too significant for real-time applications or rapid design explorations. Our approach falls into the same category of using mesh representation for the input geometry, however, we use mesh-based graph neural networks to reduce online computation time significantly.

**Simulation of Anisotropic Materials** Realistic simulating of many phenomena in nature must take into account their inherent material anisotropy, *e.g.* muscle deformation (26), plant biomechanics (27), material fracture (28), *etc*. Within the scope of this work, we focus on simulating deformable objects within the hyperelastic regime. Within this realm of research, many forms of anisotropic energies have been extensively studied, for instance, transverse isotropic elasticity (29; 30; 31), orthotropic elasticity (32), and generalized anisotropic elasticity (33; 34). We focus on transverse isotropic elastic material where a base isotropic material is augmented with freely oriented fibers to achieve directional-dependent properties. This allows for easy integration into existing isotropic formulations. While anisotropic material properties have been extensively studied for mesh-based simulation, representing directional-dependent behavior with neural representation remains unexplored. We identify a key limitation factor for existing mesh-based neural representations and propose a simple yet effective strategy for better capturing material anisotropy.

**Neural Representation** Deep neural representations hold substantial promise as alternatives for modeling complex physical systems while significantly reducing computational requirements when compared to conventional approaches (35; 36; 37). One stream of research relies on ground-truth simulation data for learning surrogate models, *e.g.* for fluid dynamics (38), character animation (39), and modeling nonlinear material properties (40; 41). With the advancement of physics-informed learning (42), another line of research leverages physical laws directly as loss functions to enable self-supervised learning (43; 44; 45; 46). In this manner, neural networks learn not only from existing data but also from the inherent physics governing the system. We also opt for an unsupervised training strategy where the variational formulation of the physics laws directly as loss functions.

However, to the best of our knowledge, our work is the first to explore material anisotropy for neural representations of deformable solids with graph neural networks.

**Mesh-based Graph Neural Networks**   Recent advancements in graph-based neural network architectures (47; 48; 49) offer a new paradigm for soft-body simulations (2; 50; 51). Specifically, mesh graph networks have emerged as a promising alternative to conventional finite element methods for simulating, for instance, fluid (52; 53), solid (54; 55; 2), cloth (3), *etc*. Unlike grid-based methods (56; 57; 35), their unstructured nature allows for easy generation of simulation domains and resolutions. Most related to our approach is MeshGraphNet (2), where an encoder-processor-decoder network architecture is leveraged to predict accelerations per time step. While their approach is able to capture a range of phenomena governed by physics PDEs, challenges remain for material anisotropy. We propose a novel and easy-to-implement edge feature decomposition operation to encode directional information during training. As we demonstrate in the result section, this modification significantly improves the performance of learning anisotropic material properties.

## 3   METHOD

In this section, we describe the machinery required for training GNNs with directional encodings. Our approach builds upon an encoder-processor-decoder network architecture with a novel edge feature decomposition scheme aimed at capturing material anisotropy (Sec. 3.1). We adopt a self-supervised training paradigm and use the variational formulation for implicit Euler as the loss function 3.2. We provide sampling, training, and implementation details in Sec 3.3.

### 3.1   MODEL ARCHITECTURE

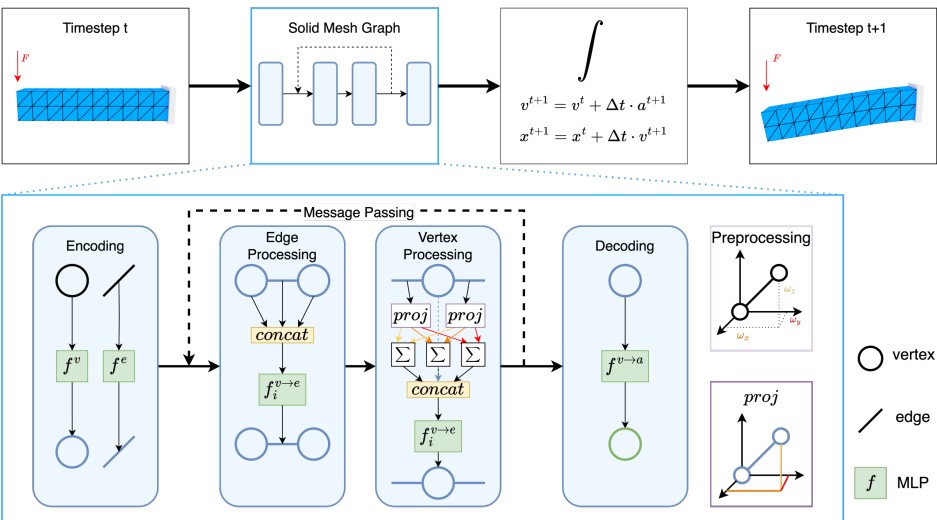

Figure 2: Pipeline. Our method takes the current states of a deformable object and its boundary conditions as input and predicts end-of-time-step accelerations using a graph neural network. These accelerations are then used to obtain the deformed state for the next time step (first row). We leverage an encoder-processor-decoder architecture and propose a novel edge decomposition operation to encode directional information during message passing (second row).

We define the simulation mesh as a graph $G = (\mathbf{V}, \mathbf{E})$ with nodes $\mathbf{V}$ and edges $\mathbf{E}$. Each node is associated with a coordinate vector $\mathbf{x}$ and additional physical parameters such as mass, external forces, and Dirichlet boundary conditions. We refer to these parameters as vertex features $\mathbf{v}$. Likewise, we use $\mathbf{e}$ to denote edge features, which include relative vertex positions and fiber orientations.

Our neural representation builds on the encode-process-decode architecture (48), where two distinct multilayer perceptrons (MLPs) are used as encoders to extract vertex and edge features. The encoded

features are then processed with a set of MLPs during a fixed number $L$ of message passing steps. In each step, all edge and vertex features are processed using the same MLPs, but each step has its own vertex and edge MLP. Finally, a decoder MLP is used to transform vertex features to end-of-time-step accelerations. The predicted accelerations are used to update vertex positions. See Figure 2 for an overview.

**Encoding and Decoding**  Our encoder and decoder MLPs largely follow MeshGraphNets (2). The input vertex and edge features are transformed into latent feature vectors through encoder MLPs $f_v$ and $f_e$,

$$\tilde{\mathbf{v}} = f_v(\mathbf{v}), \quad \tilde{\mathbf{e}} = f_e(\mathbf{e}), \tag{1}$$

where $\tilde{\mathbf{v}}$ and $\tilde{\mathbf{e}}$ denote updated feature vectors. The vertex decoder $f_{v \to a}$ maps vertex features to end-of-time-step accelerations for a given vertex,

$$\mathbf{a} = f_{v \to a}(\mathbf{v}) \,, \tag{2}$$

which are then used to compute end-of-step positions.

**Direction-aware Message Passing**  Our key contribution lies in the message passing step where we leverage directional encodings to better preserve information on anisotropic states of deformation. Specifically, we update per vertex and edge feature as

$$\tilde{\mathbf{e}} = \mathbf{e} + f_{v \to e}(\mathbf{e}, \mathbf{v}_0, \mathbf{v}_1),$$
$$\tilde{\mathbf{v}} = \mathbf{v} + f_{e \to v}(\mathbf{v}, \sum_{\mathbf{e}_j \in \mathcal{N}_i} \omega_{x,j} \mathbf{e}_j, \sum_{\mathbf{e}_j \in \mathcal{N}_i} \omega_{y,j} \mathbf{e}_j, \sum_{\mathbf{e}_j \in \mathcal{N}_i} \omega_{z,j} \mathbf{e}_j), \tag{3}$$

where $\mathbf{v}_0$ and $\mathbf{v}_1$ are the vertex features for the two endpoints of a given edge, $\mathbf{e}_j$ loops over the features for all edges incident to vertex $i$, and $f_{v \to e}$ and $f_{e \to v}$ are the edge and vertex processor MLPs respectively. We use $+$ to denote residual connections (58).

It is important to note that MeshGraphNet (2) aggregates edge features directly to update vertex features. This operation, however, does not distinguish deformations in different directions. To understand this problem, consider a mesh edge that is oriented along the $x$-axis in material space. Since the edge stores relative position between its endpoints, it cannot sense deformation along the $y$- and $z$-directions, which leave relative positions along the $x$-axis unchanged. Nevertheless, the feature aggregation scheme used in MeshGraphNets does not consider this dependence of sensing capacity on edge orientation, which ultimately limits its ability to capture directional deformation and model material anisotropy. By contrast, our novel encoding scheme projects mesh edges onto an orthonormal material-space basis such as to measure their capacity to sense deformation along different coordinate axes. The resulting coordinates are then used to decompose the original edge feature into three weighted components that are averaged individually. Using this directional encoding, an edge that aligns well with a given direction of deformation is given more authority to determine the averaged feature than an edge that is almost orthogonal to that direction. As a result, our method is able to preserve directional deformation during message passing and can thus better model anisotropic materials. We note that the edge weights $\omega_{x,j}$, $\omega_{y,j}$, and $\omega_{z,j}$ are computed from the rest state edge vectors and remain constant during training. Concretely, the weights for a given edge $\mathbf{E}_j$ are computed as

$$\omega_{x,j} = \frac{\mathbf{E}_j}{||\mathbf{E}_j||} \cdot \mathbb{E}_x, \quad \omega_{y,j} = \frac{\mathbf{E}_j}{||\mathbf{E}_j||} \cdot \mathbb{E}_y, \quad \omega_{z,j} = \frac{\mathbf{E}_j}{||\mathbf{E}_j||} \cdot \mathbb{E}_z, \tag{4}$$

where $\mathbb{E}_x$, $\mathbb{E}_y$ and $\mathbb{E}_z$ are unit-length basis vectors. We further note that this modification requires minimal changes to standard mesh-based graph neural network architectures, allowing for easy integration of our approach into an existing framework. As we demonstrate in the result section, our directional feature encoding scheme leads to significantly improved performance for learning material anisotropy.

## 3.2 Physics-based Loss Function

**Spatial Discretization**  We resort to tetrahedral finite elements with linear basis functions to model the nonlinear dynamics of deformable solids. Our network operates on the edges and nodes of the simulation mesh and performs message passing on the corresponding graph. Adhering to standard finite element practice, our loss functions by summation of per-element potentials.

**Loss Function** To allow for efficient self-supervised learning, we formulate our loss function to directly penalize the violation of the dynamic equilibrium conditions. To enable robust time stepping for larger step sizes, we use backward Euler integration, *i.e.*, a first-order accurate implicit time stepping scheme (59). Instead of directly solving the resulting system of nonlinear equations, we follow the variational formulation of Martin *et al.* (17) and convert the root finding problem into an energy minimization problem. We use the corresponding incremental potential as our physics-based loss function during training. Defining end-of-time-step positions and accelerations as $\mathbf{x}^{t+1}$ and $\mathbf{a}^{t+1}$, our total loss function reads

$$\mathcal{L}_{\text{total}}(\mathbf{a}^{t+1}, \mathbf{x}^{t+1}) = \mathcal{L}_{\text{elastic}}(\mathbf{x}^{t+1}) + \mathcal{L}_{\text{external}}(\mathbf{x}^{t+1}) + \mathcal{L}_{\text{kinetic}}(\mathbf{a}^{t+1}) . \tag{5}$$

Our $\mathcal{L}_{\text{elastic}}$ term captures the elastic energies for both isotropic and anisotropic deformation. We focus on transversely isotropic materials, where anisotropic fibers are embedded in an isotropic base material. Such materials are widely used for physics-based modeling of, *e.g.*, fiber-reinforced composites, and biological tissue. We adopt the widely used Saint Venant–Kirchhoff model (60) for the isotropic base material and augment it with an anisotropic term that models the effect of embedded fibers with a given orientation. The elastic energy for a given tetrahedron element is defined as

$$\mathcal{L}_{\text{elastic}}(\mathbf{x}^{t+1}) = \bar{v} \left( \frac{\lambda}{2} (tr(\mathbf{E}))^2 + \mu tr(\mathbf{E}^2) + \kappa (\mathbf{d}^\mathsf{T} \mathbf{F}^\mathsf{T} \mathbf{F} \mathbf{d} - 1)^2 \right) , \tag{6}$$

where $\mathbf{E} = \frac{1}{2}(\mathbf{F}^\mathsf{T}\mathbf{F} - \mathbf{I})$ is the nonlinear Green strain, $\mathbf{F}$ is the deformation gradient, and $\mathbf{d}$ is the fiber direction. Furthermore, $\bar{v}$ is the undeformed volume of an element, and $\lambda$ and $\mu$ are Lamé parameters for defining the material properties. Finally, $\kappa$ is the Young's modulus for fiber stiffness.

The kinetic energy term is defined as

$$\mathcal{L}_{\text{kinetic}}(\mathbf{a}^{t+1}) = \frac{1}{2} (\Delta\mathbf{v}^{t+1})^\mathsf{T} (\Delta\mathbf{v}^{t+1} \odot \mathbf{m}_v), \tag{7}$$

where $\Delta t$ is the simulation time step size, $\Delta\mathbf{v}^{t+1} = \Delta t \mathbf{a}^{t+1}$ are velocity increments and $\odot$ denotes element-wise vector-vector multiplication between the velocity increments and masses for all vertices within an element.

We further define the external energy corresponding to the work done by external loads as

$$\mathcal{L}_{\text{external}}(\mathbf{x}^{t+1}) = \mathbf{f}_{\text{ext}} \cdot \mathbf{x}^{t+1}, \tag{8}$$

where $\mathbf{f}_{\text{ext}}$ is a vector containing all external forces.

### 3.3 TRAINING AND IMPLEMENTATION DETAILS

**Sample Generation** We generate our training samples using combinations of simple geometries, *e.g.* rectangular and cylindrical beams (36 in total) with different mesh topologies and resolutions. The training mesh resolution is between 60 to 120 elements. We uniformly sample the force direction and magnitude $(0 - 10kN/m^3)$ applied to each mesh element. For non-uniform loading scenarios, we add additional forces to each element with a probability of $5\%$. Magnitude and direction are randomly sampled for each element $(0 - 15kN/m^3)$. Finally, we include traction samples with a probability of $10\%$ with fixed direction in $+z$ axis and amplitudes between $0 - 100kN/m^3$. For material anisotropy, we uniformly sample fiber orientations and magnitudes between $0 - 10E$ where $E$ is the Young's Modulus of the base material which is fixed to be $100kPa$. To increase stability for long-time inference rollouts, we find it crucial to sample not only undeformed states with random forces but also deformed states with non-zero kinetic and elastic energies. We apply the above parameter sampling procedure to these pre-deformed samples as well.

**Training** Our framework is implemented in C++ using LibTorch. We use the Adam (61) optimizer with a learning rate of $5 \times 10^{-5}$ and a weight decay rate of $10^{-4}$ per one hundred iterations. Each training sample is unique and randomly sampled. The batch size is set to 1 and we train a total of $672,000$ epochs. All of our MLPs have two hidden layers with 128 neurons per layer and SiLU activation functions (62). Layer normalization is applied to all layers except the final decoder MLP. All input features except the one-hot encoded anchored vertices are normalized. The encoder MLPs produce output features of size 128, while the vertex decoder yields features of size 3. Following MeshGraphNets, we perform 15 message-passing steps.

Vertex input features consist of a one-hot-ended vector containing Dirichlet boundary conditions, vertex velocities, vertex mass and vertex external forces. Edge input features consist of two vectors containing the edge direction of both undeformed configuration and current deformation. Both vectors are normalized and their norm is added as a separate feature. Additionally, all edges contain another vector with fiber direction and magnitude.

During training, we introduce perturbations to both nodal velocity and positions using zero-mean noise. The variance for velocities is stochastically sampled from the range of $0-5\times10^{-2}m/s$, while the variance for position noise falls within the interval of $0-10^{-3}m$. This perturbation process, akin to MeshGraphNets, plays a pivotal role in ensuring the stability of the neural network for long rollouts. The network is trained on a workstation with an *AMD Ryzen 7 5800X* CPU and an *NVIDIA GeForce RTX 3080Ti* GPU. Training takes around 5 days, whereas inference takes $9ms$ for a mesh of 100 elements.

The Lamé parameters are computed from Young's modulus (100kPa) and Poisson's ratio (0.48) of a soft rubber-like material. When performing time stepping, we use a step size $\Delta t$ of $0.02s$. We will release our code upon acceptance.

## 4 RESULTS

In this section, we compare our results to the state-of-the-art mesh graph neural network, Mesh-GraphNets (2) on a set of qualitative and quantitative experiments. Since MeshGraphNets are trained in a supervised fashion, for fair comparisons, we implemented an unsupervised version using their network architectures with only modifications to the loss function to accommodate self-supervised learning. We demonstrate that our approach outperforms this baseline in terms of convergence speed, the ability to capture material anisotropy, and volume preservation for nearly incompressible materials. We further use a standard finite element solver to generate ground truth data for reference.

**Convergence** We begin by comparing our approach with MeshGraphNets for different numbers of test rollouts (Figure 3). We generate 15 random configurations, *i.e.* different mesh topologies, force magnitudes, and directions, as test sets for all approaches and compute the difference in energy with respect to the ground truth value obtained from our reference simulation. After each training iteration, we evaluate all networks on the same test sets for different numbers of rollout steps in order to gauge their stability for sequences of different lengths. In particular, longer rollouts are useful to test whether predictions are converging toward static equilibrium. As can be seen from Figure 3, our approach consistently improves on MeshGraphNets, showing substantially faster convergence in all cases. It can also be noted that our method converges to equilibrium states with lower total energy.

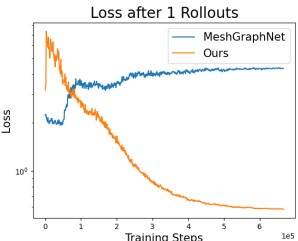 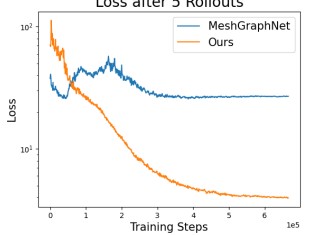 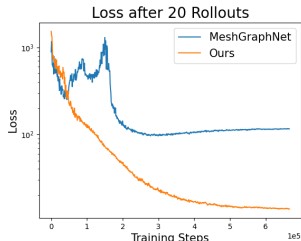

Figure 3: Network convergence. We compare the convergence behavior for our approach with MeshGraphNets on a test set for different rollout lengths. As can be seen in these figures, our approach converges to lower energy states much faster while remaining stable for longer horizons.

We attribute the significant discrepancy of MeshGraphNets to its limited ability to capture the anisotropic fibers. To verify this hypothesis, we visualize the energy difference to ground truth data for the fiber term and the sum of all terms separately. In this example (Figure 4), a beam with fibers along its long axis is loaded along the fiber direction. As can be seen from the plot shown to the left, the error in the fiber term for MeshGraphNets dominates the overall energy profile, leading to 10 times larger error compared to our method.

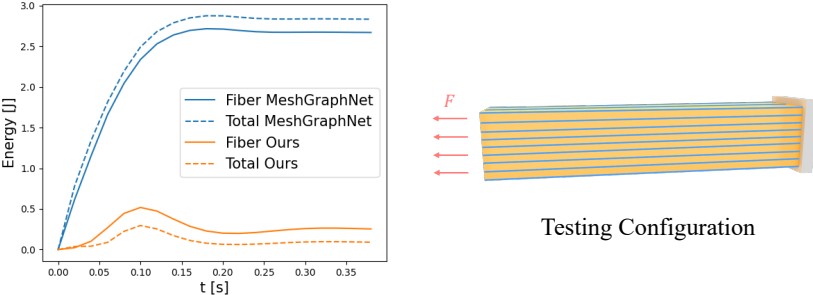

Figure 4: Fiber and total energy error. A beam under uni-axial tension with fibers aligned with the direction of loading (*right*). We report the fiber and total energy error compared to simulation references (*left*). Due to the limited capability of capturing material anisotropy, the error from the fiber term dominates the overall error leading to significant deviation from ground truth data. Our approach, on the other hand, demonstrates $10$ times higher accuracy.

**Anisotropic Elasticity**    To quantify the difference in terms of capturing anisotropic elasticity, we compare our approach with MeshGraphNet on a set of uniaxial loading test cases with fiber reinforcements in different magnitudes and directions (see Figure 5). When fiber reinforcements are collinear with the loading direction, they introduce strong resistant forces upon tensioning. Consequently, larger stress magnitudes for a given strain rate. As can be seen in the slopes of the curves in Figure 5 (a,b), our model successfully captures this highly anisotropic behavior for different fiber stiffness whereas MeshGraphNets leads to poor matching behavior. Note that for strong fibers, the predictions from MeshGraphNets deviate already for small strain. When fibers are aligned orthogonal to the loading directions, they have minimal effects on the directional stress magnitude. This behavior is again captured by our model (Figure 5 (c)).

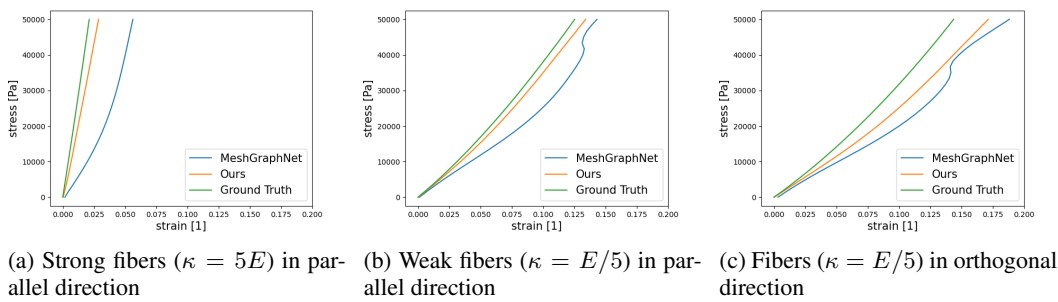

(a) Strong fibers ($\kappa = 5E$) in parallel direction

(b) Weak fibers ($\kappa = E/5$) in parallel direction

(c) Fibers ($\kappa = E/5$) in orthogonal direction

Figure 5: Strain-stress curves. We compare our approach with MeshGraphNets on a set of uniaxial loading cases with different fiber orientations and magnitudes. We use $E$ to denote Young's modulus of the base material. The predictions from our approach track the ground truth solution consistently better than MeshGraphNets and do not suffer from instabilities for larger strain rates.

**Volume Preservation**    In addition to capturing explicit material anisotropy, direction encodings also facilitate learning volumetric effects pertaining to the Poisson ratio, *i.e.* when a tension load is applied in one direction, causing the orthogonal directions to contract in order to preserve the material volume. In this experiment, we compare our approach and MeshGraphNets to the reference simulation on volume preservation of a beam under a constant tensile force. We report the maximum relative percentage error over all elements in Figure 6. As can be seen from this plot, MeshGraphNets leads to volume change up to $60\%$ whereas our approach exhibits almost zero volume changes.

**Tip Displacements**    Complementing previous examples where tension modes are examined, we now shift to bending modes for more analysis. In this example, we quantitatively validate our approach by comparing the tip displacement error for a cantilever beam to its reference simulation.

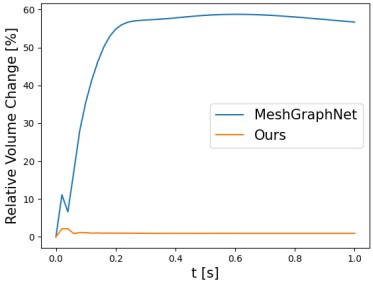

Figure 6: Volume preservation error. We plot the maximum relative percentage error for all elements in a deformed beam under tension. While our directional feature encoding leads to almost zero volume change compared to the simulation baseline, MeshGraphNets permits volume changes up to 60%.

We consider two extreme testing scenarios for fiber orientations, one that is aligned with gravity in its rest shape (more deformation) and one that is orthogonal to it (less deformation). They are referred to as *parallel* and *orthogonal* in Table 1. We test all approaches with two beam topologies, namely rectangular (row *1-6*) and cylindrical beams (row *7-12*). For this set of experiments, we use the same stiffness for both the base material and the fibers. As reported in Table 1, our approach consistently outperforms the baseline method in terms of accuracy across all tested scenarios and beam topologies.

| Fiber Orientation | Beam Topology | Method | Tip Displacement Error($m$) |
|---|---|---|---|
| parallel | rectangular | MeshGraphNets | 0.0399 |
| parallel | rectangular | **ours** | **0.0119** |
| orthogonal | rectangular | MeshGraphNets | 0.0902 |
| orthogonal | rectangular | ours | **0.0510** |
| parallel | cylindrical | MeshGraphNets | 0.1111 |
| parallel | cylindrical | **ours** | **0.0776** |
| orthogonal | cylindrical | MeshGraphNets | 0.1400 |
| orthogonal | cylindrical | **ours** | **0.0977** |

Table 1: Tip displacement comparisons. We consider two types of beam structures under gravitational force with one end of the beams fixed and leaving the other end free. Reinforcement fibers are set to be either parallel or orthogonal to gravity. As can be seen from the tip displacement errors reported, our approach demonstrates significantly higher accuracy compared to MeshGraphNets.

**Imbalanced Forces** In this experiment, we consider the physically imbalanced force in the configuration generated by MeshGraphNets and our approach. The gradient of our loss function w.r.t. nodal positions amounts to the force equilibrium condition governed by Newton's second law of motion, which should vanish at stable configurations. We therefore refer to the nonzero gradients as imbalanced forces. In Table 2, we report the imbalanced force magnitude from network predictions for a cantilever beam under static force equilibrium configurations. Same as in the previous example, we use the same stiffness for both the base material and fiber reinforcements. We apply a force density with its direction with gravity with two magnitudes ($1000N/m^3$ and $5000N/m^3$). The fiber directions are varied from 45 to 90 degrees with 90 being orthogonal to the force direction. As can be seen from the statistics for average and maximum nodal imbalance forces, our approach reduces the mean error by $80\%$ on average and the maximum error up to $90\%$.

**Generalization** Finally, we demonstrate that our network generalizes to unseen geometries with different fiber layouts (Figure 7). In the first example, we add fibers to a T-shaped deformable object to resist bending load whereas in the second one, the fibers resist compression force for a Y-shape geometry. The applied forces and fiber orientations are shown in the insets.

| Fiber Direction | Force Density ($N/m^3$) | Method | Imbalanced Force ($N$) Max/Mean |
|---|---|---|---|
| 45° | 5000 | MeshGraphNets | 77.84 / 16.71 |
| 45° | 5000 | **ours** | **14.01 / 3.747** |
| 45° | 1000 | MeshGraphNets | 46.40 / 12.20 |
| 45° | 1000 | **ours** | **4.321 / 1.446** |
| 60° | 5000 | MeshGraphNets | 76.36 / 16.83 |
| 60° | 5000 | **ours** | **18.92 / 4.205** |
| 60° | 1000 | MeshGraphNets | 42.98 / 12.07 |
| 60° | 1000 | **ours** | **4.760 / 1.608** |
| 90° | 5000 | MeshGraphNets | 67.89 / 16.25 |
| 90° | 5000 | **ours** | **18.78 / 4.447** |
| 90° | 1000 | MeshGraphNets | 44.69 / 11.86 |
| 90° | 1000 | **ours** | **4.105 / 1.841** |

Table 2: Physically imbalanced force. We compare the physically imbalanced force in the predictions from MeshGraphNets and our approach for different fiber orientations and force densities. Our approach significantly reduces both the average and the peak error.

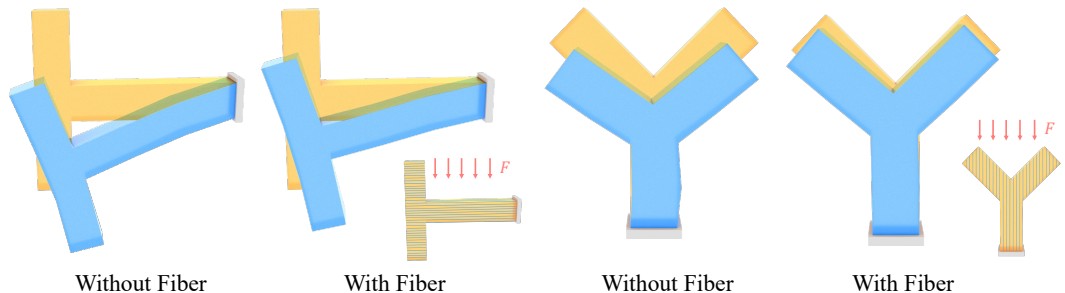

Without Fiber      With Fiber      Without Fiber      With Fiber

Figure 7: Network generalization. We apply our approach to geometries significantly different from our training set. As can be seen from these two examples, the embedded reinforcement fibers play a crucial role in determining the deformed configurations. This material anisotropy is faithfully captured by our approach. The rest and deformed states are shown in orange and blue respectively.

## 5   CONCLUSION

We have presented a novel mesh-based graph neural network architecture for learning the elastodynamics of anisotropic elastic materials. Whereas state-of-the-art approaches are limited to isotropic materials, we propose a novel and easy-to-implement edge feature decomposition scheme that preserves directional information during message passing and thus allows for the modeling of material anisotropies. We demonstrate on a set of qualitative and quantitative examples that our approach outperforms the state-of-the-art method by significant margins. Although we focus on nonlinear elasticity in this work, we believe that our feature decomposition scheme can benefit other applications of graph neural networks that involve direction-dependent behavior.

### 5.1   LIMITATION AND FUTURE WORK

While our approach generalizes well to unseen meshes with similar resolution, we would like to leverage hierarchical representations (51; 50) to apply our approach across a wider range of mesh resolutions. Another interesting avenue for future research is to leverage our neural representation as an efficient and smooth surrogate model for inverse design tasks, *e.g.* shape optimization, where analytical derivatives can be easily obtained through auto differentiation of the network. Finally, our current formulation enables efficient self-supervised learning of anisotropic material properties through physics-based training losses. In the future, we would like to include measurements from real data to obtain neural representations of fiber-reinforced mechanical metamaterials.

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
