# OpenReview forum: "Graph Neural Networks with Directional Encodings for Anisotropic Elasticity"
_ICLR.cc/2024/Conference — Submitted to ICLR 2024_

### Official Review · Reviewer_EyHC · 2023-10-23

**Soundness:** 3 good
**Presentation:** 2 fair
**Contribution:** 2 fair
**Rating:** 5
**Confidence:** 4

**Summary:**

This paper proposes directional encodings for edge features in GNN to help extracting the directional information during message passing. The proposed method outperforms the MeshGraphNet as shown in the experiments.

**Strengths:**

* The proposed directional encodings are easy to integrate with GNNs.
* The proposed method obtain superior performance over MeshGraphNet.

**Weaknesses:**

This paper seems technically limted and compared with only one basic GNN. Some weaknesses are as follows:
1. Only the basic MeshGraphNet is compared. For example, Is HOOD (CVPR'23) able to achieve better performance?
2. The equation 3 has a similar format of attention with 3 heads. What's the performance of Graph Attention Network (GAT)? Will attention scores be able to replace the weights and become a more general format of equation 3?
3. The simulated objects in datasets seem only consist of limited number of elements (60-120). Note that MeshGraphNet is able to deal with thousands of particles.  Is this method able to handle cases with more elements?

**Questions:**

1. Does the "5 rollouts" in Fig.3 mean the model autoregressively predicts 5 steps?
2. Is there any video result of the process of the deformations?

---

> ### Author Response · Authors · 2023-11-14
>
> Thank you for your feedback and for acknowledging that our approach achieves superior performance over the state-of-the-art method.
>
> __Higher resolution__
>
> Previous work on hierarchical representations has shown promising results for GNNs when extending to higher resolution meshes. Our work focuses on an orthogonal question, i.e. how to extend MeshGraphNets from isotropic materials to anisotropic materials? While our solution to this question would likely benefit from them, hierarchical representations alone cannot help with anisotropy.
>
> __Comparison to HOOD__
>
> HOOD employs MeshGraphNets in a cloth simulation setting, incorporating a hierarchical representation for accelerating information propagation during message passing. In the absence of a hierarchical structure, their network closely resembles MeshGraphNets and is thus limited to isotropic materials. Our approach, on the other hand, extends MeshGraphNets to anisotropic material.
>
> __GAT__
>
> Although attention mechanisms might offer some assistance, our approach introduces a simpler yet highly effective strategy. While attention masks must be learned during training, our method only requires no extra mechanism. Our modifications incur minimal additional cost for integrating into an existing Graph Neural Network (GNN) architecture.
>
> __Questions__
>
> Yes “5 rollouts” means using the model to autoregressively predict 5 steps. This plot is meant to show that the network is stable for longer rollouts. We will include animations in the supplementary video for the final version.
>
> We hope these answers provide sufficient reasons to raise the score. Otherwise, please let us know what are the remaining questions that are holding you back.

---

> > ### Comment · Reviewer_EyHC · 2023-12-05
> > **Official Comment by Reviewer EyHC**
> >
> > Thank you for your reply.  However, I'm not fully convinced and would still maintain the original scores.
> >
> > The main reason that I asked about the attention and hierarchical structure is that the exisitng methods are sufficient to handle the anisotropy. However, no comparison with them is provided and no evidence can show that the proposed method is more effective than them. Simply comparing with the naive baseline model is not enough to me.

---

### Official Review · Reviewer_kvP8 · 2023-10-30

**Soundness:** 2 fair
**Presentation:** 3 good
**Contribution:** 2 fair
**Rating:** 3
**Confidence:** 3

**Summary:**

This paper proposes a graph network for simulating deformable solids with anisotropic hyperelastic materials. The key contribution lies in its treatment of material anisotropicity in the network module. The paper compares its results with MeshGraphNet, a standard GNN baseline for deformable solid simulation.

**Strengths:**

At a very high level, the core idea in the paper is easy to follow for people familiar with GNNs and deformable solid simulation. Modeling anisotropic materials with neural networks is also an interesting problem with many potential applications.

**Weaknesses:**

Most of my concerns are reflected in the questions section below.

**Novelty and contributions**
The scope of the problem setup is narrow (anisotropic hyperelastic material without considering plasticity, contact, or collision). Therefore, I expect an in-depth study of this problem to justify its publication. This can include deep insight into anisotropic materials, comparisons with strong and fine-tuned baselines, comprehensive analysis of its generalizability, or demonstrating its exciting downstream applications with anisotropic materials.

**Methods**
The notations in the technical methods are confusing, and I feel it would be challenging for people lacking deformable solid simulation background to understand and reproduce this paper. I left a number of specific questions in the “Questions” section below.

**Experiments**
- I feel the problem size is too small to conduct meaningful analysis on GNNs. 60-120 elements (the training problem size) are considered very few for deformable solid simulation and can be solved quickly and accurately with numerical methods. The “DeformablePlate” in MeshGraphNet contains ~1200 nodes (so >= 300 tetrahedrons, with contact/collision handling). A modern, GPU-based numerical simulator can probably scale this up even more without losing speed or accuracy if no contact/collision needs to be solved, which seems to be the setup of this paper. I do agree with the intro that learning-based approaches can “strike a balance between accuracy and efficiency,” but such a tradeoff doesn’t need to exist for very small problems.

- Baselines: The performance of the baseline seems much worse than what the original MeshGraphNet paper reported. A concrete example is Fig. 6: if MeshGraphNet caused 60% volume change, their “DeformablePlate” example would have exhibited very obvious artifacts. I wonder whether the variational loss + MeshGraphNet combination negatively influenced its performance, but I am not sure. Also, following my first point, I feel the paper lacks a crucial comparison to the reference simulator in terms of speed and accuracy.

- I also feel the study on the method’s generalizability is limited. Having more diverse, spatially varying fiber orientations other than horizontal and vertical ones would be more convincing. Testing it on more realistic hyperelastic materials (e.g., Neohookean or corotated) would be useful as well.

**Questions:**

**Technical questions about the network design**
- I suspect “f^{v->e}” in “vertex processing” of Figure 2 should be replaced with “f^{e->v}”.
- I don’t quite follow why fiber orientation is an edge feature. It seems more reasonable to consider fiber orientation as a (finite) element feature if there is such a thing in the network module. The reason why I have this feeling is that F, the deformation gradient, is typically an element quantity in linear tetrahedron finite elements, and Eqn. (6) indicates that it’s convenient to consider d as an element quantity as well.
- Could you clarify the notation “E_j” in Eqn. (4)?
- How are E_x, E_y, E_z in Eqn. (4) defined? I am guessing that one of them is d and the other two are orthogonal to d, but I am not quite sure.
- I am not sure I get the intuition behind explicitly formulating three weighted sums of e_j in  Eqn. (3). Part of me wonders whether it is truly necessary, as the directional information d is already provided in the edge feature. I can accept that doing so does not hurt, but it would be nice if the authors could provide more insights into this design decision. This seems crucial for the paper’s technical contribution, so I want to make sure I fully understand it.

**About the loss function**
- Eqn. (5): Is x^{t+1} computed from a^{t+1} or is it an independent variable? I am guessing that the network produces a^{t+1}, which is then used to compute x^{t+1}, and both a^{t+1} and x^{t+1} are fed into this loss function.
- Eqn. (6): I understand that the text already mentioned that the loss function sums up per-element potentials, but I’d still appreciate a more rigorous writing of the strain energy, i.e., adding a proper sum over all finite elements and defining how F is computed from x^{t+1} (e.g., by citing 16 or similar literature). Echoing my question above, I’d also like to understand how dFFd is computed in a single element.
- While I appreciate the choice of using the incremental potential as the loss function L, I feel there are some subtleties after incorporating the network. Let x be the new position and theta be the network parameters. A stationary point of min_x L(x) nicely solves implicit Euler integration because L is its variational form, but a stationary point of min_theta L(x(theta)) only satisfies dL/dtheta = dL/dx * dx/dtheta = 0. From a theoretical perspective, whether this guarantees dL/dx = 0 (the true solution to the implicit Euler integration) is not obvious to me.

**About network training**
- Could you elaborate on how the fiber orientations are uniformly sampled in each element? Also, is each element assigned an independently sampled direction? The results seem to contain only horizontal and vertical directions shared by all anisotropic elements.

**About the “Convergence” experiment**
- Is Fig. 3 displaying the training loss or its difference from the ground truth incremental potential solved by the reference simulation?
- I suggest adding another figure that directly visualizes the difference between the network-predicted x^{t+1} and the reference x^{t+1} from the numerical simulator.
- Fig. 4: Why is the “total” energy difference lower than the “fiber” term difference?
- For both Figs. 3 and 4, I am not sure whether the worse performance of MeshGraphNet should be attributed to the network lacking the direction-aware message-passing mechanism or the decision to use the new loss function in MeshGraphNet.

**About the “Volume Preservation” experiment**
- A minor comment is that Poisson’s ratio = 0.48 does not mean zero volume change. It would be informative to add a third curve in Fig. 6 showing the volume change from the reference simulator.

**About the “Tip Displacement” experiment**
- I am trying to understand the significance of the “Tip Displacement Error.” What is the average size of the finite elements in these scenes?
- Also, how many elements does a test scene typically have?

**About the “Imbalanced Forces” experiment**
- I like this experiment more than the others, but again, what is the size of the finite elements? Without knowing it, I was having a difficult time calibrating the “Force Density” column and the “Imbalanced Force” column in Table 2.

**About the “Generalization” experiment**
- I didn’t find quantitative data about this experiment. In particular, how different is it from the deformed shape computed from the reference simulator?
- I am curious to see the generalization to more diverse, spatially varying fiber orientations.

---

> ### Author Response · Authors · 2023-11-14
>
> Thank you for your feedback.
>
> __Unsupervised training for MeshGraphNets__
>
> To test whether the sub-par performance of MeshGraphNets might be due to our unsupervised learning approach (the original work used supervised training), we trained MeshGraphNets with only isotropic material using unsupervised learning.
> We observe that MeshGraphNets performs significantly better for isotropic materials - see results in this anonymous [link](https://i.postimg.cc/xTFRWRyf/image-259.png). These results suggest that the sub-par performance of MeshGraphNets for anisotropic materials is indeed due to its inability to capture anisotropic behavior, not to the way in which it is trained.
>
>
> __Contact and higher resolution__
>
> We refer to the [Contact, higher resolution, complex scene] response to R2-r4js
> Regarding performance gains, we reported in the training paragraph of section 3.3 that the inference of our network on a mesh of 100 elements takes only 9ms. The same examples are around 10 to 20 times slower in our reference FEM simulation. MeshGraphNet reports two to three orders of magnitude performance gain over the reference FEM solver solver. Since our network architecture and hyperparameters follow MeshGraphNet, we expect the same performance when scaling to higher resolutions.
>
> __NeoHookean & Corotated__
>
> We agree that NeoHookean and the corotated constitutive models are viable options. However, these models are still isotropic and would need to be augmented with stiffening fibers in the same way as we do with our StVK-based material. Although we believe that our method readily extends to different isotropic base materials (NeoHookean, Mooney-Rivlin, etc.), our focus is on anisotropy and we therefore opted for a simple StVK model.
>
> __Technical Question__
>
> We generally think of fiber directions as continuous vector fields. These fields can be sampled at vertices, edge midpoints, or element centroids. One representation could be converted into any other using simple interpolation. We agree that fiber orientation could be an element feature, but we chose the edge representation for simplicity (edges and vertices are modeled in the MeshGraphNets graphs whereas elements are not).
>
> E_j denotes an edge in the undeformed configuration, i.e. E_j = x_k - x_l, where x_k and x_l are mesh vertices. E_x, E_y, E_z are Cartesian basis vectors, [1, 0, 0], [0, 1, 0] and [0, 0, 1]. These basis vectors are decoupled from fiber orientations, which are randomly oriented during training and are in general not aligned with E_x, E_y, and E_z. We opt for this projection operation to decouple information in different directions. As can be seen in all our experiments, we observe significant performance gains with this approach.
>
> We will fix the typo in Figure 2.
>
>
> __Loss function__
> Thanks for pointing these out. Yes, x^{t+1} should be a function of a^{t+1}. Computing deformation gradient follows standard finite element theory for linear tetrahedral elements. We will include the computation in the supplementary material. The first-order optimality condition with the network in the loop is indeed that the gradient w.r.t. network weights should be zero. We will clarify this in our final version.
>
> __Volume Change__
>
> We agree that a Poisson ratio of 0.48 does not imply volume preservation, which is why we are plotting the relative percentage error with respect to the ground truth simulation. This description is indeed in the caption but we are happy to make it clearer.
>
> __Fiber directions__
>
> Different fiber orientations are evaluated in Table 2 when examining force imbalance. In the remaining examples, we specifically opt for fiber directions aligned with and orthogonal to the applied forces to highlight their effects. Aligning the deformations with fibers in these directions is more intuitive for readers than scenarios involving randomly oriented fibers. We consider homogeneous fibers, i.e. same fiber directions for all elements,  with directions and magnitude randomly sampled during training.
>
> __Convergence__
>
> Fig.3 displays the difference between the network predictions on the test set and the ground truth simulation, i.e. test loss normalized by ground truth value.
>
> The energy corresponding to the external force load can be negative. Therefore, the total energy difference can be smaller than the fiber energy difference.
>
> __Element Size__
>
> The cantilever beam mesh has dimensions  0.1m * 0.1m * 0.5m, resulting in a total volume of 0.005m^3. If we distribute 100 elements within the bar, each element, on average, would have a volume of 50 cm^3 and an average edge length of approximately 7.5cm.
>
> If these answers are satisfying to you, please consider raising your score. Otherwise, we are more than happy to elaborate on other questions.

---

> > ### Comment · Reviewer_kvP8 · 2023-11-22
> > **Thank you for the response**
> >
> > Thank you for the clarification and extra experiments. I remain negative about this submission and would like to maintain my original score.

---

### Official Review · Reviewer_r4js · 2023-11-01

**Soundness:** 4 excellent
**Presentation:** 3 good
**Contribution:** 2 fair
**Rating:** 6
**Confidence:** 3

**Summary:**

This paper proposes an extension to MeshGraphNets (Pfaff et al. 2021), to account for anisotropic materials.  The primary contribution is the addition of directional encodings in the message-passing GNN, such that during updates of the vertex embeddings, the corresponding incident edge features are _weighted by directional edge weights_ prior to being concatenated.  This allows the network to additionally learn anisotropic deformation.  They also devise a self-supervised loss function based on the variational formulation of the physical laws governing the simulation.  They present comparisons with MeshGraphNets, and a ground truth FEM simulator.  Their method allows the graph networks to more faithfully learn anisotropic dynamics.

**Strengths:**

The paper addresses a useful open question within the emerging topic of learned simulators, equipping message-passing networks with anisotropic elasticity.  The solution is simple but effective, and experimental results show a meaningful improvement over the baseline method.  Overall, the writing is clear, and experiments seem reproducible.

**Weaknesses:**

The technical contribution seems potentially incremental from prior work (MeshGraphNets) -- however, this is not necessarily an issue, as the experiments are well-designed, the results are solid and finding are conclusive.

I would be interested to see additional experiments, beyond the cantilever (and cantilever-like) setup, such as simulation with collision/contact.  This is not a requirement though, as the existing experiments are quite informative.

**Questions:**

- I am curious about the "material space bases" mentioned on page 4.  Could you elaborate on how the "material space bases" are defined exactly?  How are these defined relative to global coordinates, and are they defined in a canonical way?

- Would defining the local bases in a different way change the three axis-aligned weights/coefficients that are computed during preprocessing?  In turn, would this make it difficult to learn?  (Presumably the x-axis is aligned with the edge, what about the others.). Thanks, look forward to your reply.

---

> ### Author Response · Authors · 2023-11-14
>
> Thank you for your feedback and for acknowledging that the solution is simple but effective.
>
> __Incremental__
>
> We would like to emphasize that although the solution we presented appears simple, identifying the key limiting factor and proposing a simple yet effective solution was no trivial task. We argue that, rather than a disadvantage,  algorithms that combine simplicity and effectiveness are highly desirable.
>
> __Contact and higher resolution__
>
> We agree that higher-resolution simulations with intricate contacts are an exciting direction to pursue. However, our contribution is orthogonal to these aspects. We aim to extend MeshGraphNets from isotropic to anisotropic materials.
>
>
> __Material Space Basis__
>
>  We apologize for the confusion about the material space basis, we opt for this term to denote a coordinate system in the undeformed (i.e., material) configuration of the object. In our implementation, we simply use the canonical bases, i.e. unit vectors in the x, y, and z directions. The edge vectors in the undeformed configuration are projected onto these basis vectors by simply selecting the x, y, and z components.
>
> If these answers are satisfying to you, please consider raising your score. Otherwise, we are more than happy to elaborate on other questions.

---

> > ### Comment · Reviewer_r4js · 2023-11-22
> >
> > Dear authors,
> >
> > Thank you for answering my questions.
> >
> > I disagree that demonstrating that your method holds up when evaluated on a more diverse range of scenarios is orthogonal to this work.  Especially given the simplicity of the proposed method (which I agree can absolutely be a good thing), it is therefore even more important to show its generalization capability and robustness in order to be fully convincing to a general audience, beyond the narrow scope of the prior work MeshGraphNets.
> >
> > As for the material space bases, this makes sense.  In my opinion, this section could have been more clearly explained in the paper, or perhaps illustrated with a small diagram, since this is at the heart of one of the key contributions of the paper.  I had one additional question regarding this: is this weighting scheme robust to rigid transformations of the undeformed configuration, and particularly at inference time?  For example, suppose at inference time, a mesh training sample were to be rotated by 90 degrees around an axis, changing the weights for a particular edge from [1, 0, 0] to [0, 0, 1], then input to the simulator.  Would the learned simulator produce a rotation-equivariant result?
> >
> > I maintain my original score.

---

### Official Review · Reviewer_RLHk · 2023-11-01

**Soundness:** 2 fair
**Presentation:** 3 good
**Contribution:** 2 fair
**Rating:** 3
**Confidence:** 4

**Summary:**

The paper presents a novel mesh-based graph neural network architecture for learning the elastodynamics of anisotropic elastic materials. The paper proposed a novel and easy-to-implement edge feature decomposition scheme that can be able to preserve the directional information and model the material anisotropies while the previous works focus on the isotropic materials. From the submission, there are some toy examples to demonstrate the proposed method outperforms some previous work from the qualitative and quantitative view.

**Strengths:**

- The paper has very good organization in the section and experiment design, which makes it very easy to follow and learn its core idea.
- From a technical point of view, the novelty is relative enough for the conference. The deformation of anisotropic materials is very important for engineering design, material simulation, robotics, and so on.

**Weaknesses:**

- The major one is the lack of sufficient comparisons and evaluations; there is only one alternative method as the baseline to compare and demonstrate the superiority of the proposed methods. More baselines are strongly recommended, adding to the experiments and evaluations to support the proposed methods by thorough evaluations.
- For the network architecture, the message-passing operation connects the position after encoding and decoding. If we add more blocks (including message passing, edge processing, and vertex processing), what about the performance of the proposed methods?
- For the loss function, how to determine the weight for each term and the ablation study on the different weight combinations should be evaluated.
- What about the running times, such as the deformation efficiency? failure cases.
- More results on some other complex shape or material composited object are strongly recommended; the current presented results are very simple. For the proposed method, it is very interesting to see some real examples instead of synthetic ones.

Other than that, some related works should be considered as
[1] SO (3)-invariance of informed-graph-based deep neural network for anisotropic elastoplastic materials
[2] Polyconvex anisotropic hyperelasticity with neural networks
[3] RIMD: Efficient and Flexible Deformation Representation for Data-Driven Surface Modeling

**Questions:**

The strengths of the paper lie in the comprehensive information provided, the inclusion of supplementary materials, and the thorough explanations. However, the lack of novelty, limited evaluation, and other weak issues. Although the appendix serves its purpose as a resource for implementing AGILE3D, it does not significantly contribute to the field. Considering these strengths and weaknesses, I am negative about the submission currently, but I look forward to the response to the above questions.

see weakness

---

> ### Author Response · Authors · 2023-11-14
>
> Thank you for your feedback.
>
>
> __Potential Misunderstanding in the Question section__
>
> We did not include any supplemental material or an appendix including AGILE3D. We understand that many reviewers are managing lots of submissions - is there perhaps a confusion? Did you mean to pass on other feedback?
>
> __Related Work__
>
> Thanks for pointing out these related works. After close examination, we notice that the first two works do not use Graph Neural Networks, despite the occurrence of  “graph-based” in the title of the first one. The third paper does not involve constitutive modeling or machine learning. It appears to us that these works do not have an immediate relationship to our method. Please let us know if we misunderstood.
>
> __Weights__
>
> These energy terms do not conflict with each other. Rather, they describe the total mechanical energy of the physical system. Minimizing the energy in its current form amounts to finding the equilibrium configuration of the system. Although each term involves different physical parameters (Young’s modulus, Poisson’s ratio, mass density, etc), re-scaling individual terms by arbitrary weights would mean breaking the physics.
>
> __Number of Message-passing Steps__
> Improving the learned behavior by increasing the number of message-passing steps is possible, but it comes with a trade-off—more network parameters may result in longer training times and a higher likelihood of encountering local minima. Following MeshGraphNets, we opt for 15 message-passing steps to strike a balance between accuracy and computational efficiency.
>
> __Baseline__
>
> While we only compare to one GNN-based baseline, this baseline is nonetheless state-of-the-art. Follow-up works such as HOOD [Grigoriev2023] introduce various improvements, but these extensions do not address the problem we target with our work – the inherent limitations of MeshGraphNets when modeling anisotropic materials.
>
> __Complex Structure__
> We refer to the [Contact, higher resolution, complex scene] response to R2-r4js
>
> __Real experiments__
>
> Our goal in this work is to overcome a key limitation in state-of-the-art GNN-based methods, and our results suggest that our method can learn anisotropic material behavior whereas MeshGraphNets cannot. While closing the sim-to-real gap with real experiments is an interesting direction for future work, it is orthogonal to our core contribution.
>
> __Run-time__
>
> We reported in the training paragraph of section 3.3 that the inference of our network on a mesh of 100 elements takes only 9ms. The same examples are around 10 to 20 times slower in our reference FEM simulation. MeshGraphNets reports two to three orders of magnitude performance gain over the reference FEM solver solver. Since our network architecture and hyperparameters follow MeshGraphNet, we expect the same performance when scaling to higher resolutions.

---

> > ### Comment · Reviewer_RLHk · 2023-11-22
> > **response**
> >
> > Sorry for the AGILE3D, this is my mistake, please ignore it.
> > Thank you for the clarification and extra experiments. Some of my concerns are addressed.
> > According to the response for the raised weaknesses, I am not very convinced about the reply, especially for the complex cases and evaluated for real examples, I am very curious about the performance of these cases to support the generalization of the proposed method according to the diverse cases.
> >
> > Best,

---

### Author Response · Authors · 2023-11-20

Dear Reviewers,

As the responding window is coming to a close, we would very much appreciate it if feedback could be provided based on our responses. If these answers are satisfying to you, please consider raising your score. Otherwise, we are more than happy to elaborate on other questions.

Best regards,

Authors

---

### Meta-Review · Area_Chair_KT75 · 2023-12-10

**Metareview:**

A method is proposed for simulating deformable shapes with anisotropic materials. The key strength is a technical contribution over prior graph neural network based works where direction encoding of edge features allows overcoming isotropic limitations. The key weakness is insufficient evaluation for real examples or complex shapes.

Reviewer RLHk is not convinced by the lack of real results and requires an evaluation on more complex shapes. The author feedback focuses on stating the advantage over prior work MeshGraphNets, but such examples are not included. Reviewer r4js, who is the most positive, also shares the same concern on generalization ability and robustness. Reviewer kvP8 also notes the lack of analysis relative to the limited problem setup and provides detailed feedback to improve the clarity of presented methods. Reviewer EyHC further notes the absence of comparisons to recent methods, which is not provided in the author feedback. Overall, the AC agrees with the reviewer opinions that the paper may not be accepted for ICLR. The authors are encouraged to resubmit to a future venue after incorporating the extensive reviewer feedback to show results on real data and complex structures, compare to recent works and improve the clarity of the technical presentation.

**Justification For Why Not Higher Score:**

While a novel formulation is presented to handle materials beyond isotropic assumptions, the paper can improve significantly with better evaluation on realistic examples and with more details included in the presentation.

**Justification For Why Not Lower Score:**

Not applicable.

---

### Decision · Program_Chairs · 2024-01-16

Reject